# Friends or foes? How activists and non-activists perceive and evaluate each other

**Maja Kutlaca**[1]*, **Martijn van Zomeren**[2], **Kai Epstude**[2]

**1** Department of Social Psychology, Faculty of Psychology, University of Osnabrück, Osnabrück, Germany,
**2** Department of Social Psychology, Faculty of Behavioural and Social Sciences, University of Groningen, Groningen, The Netherlands

* maja.kutlaca@uni-osnabrueck.de

**Data Availability Statement:** All datasets, syntax and materials are available: https://osf.io/72yjc/.

**Funding:** The author(s) received no specific funding for this work.

## Abstract

Little is known about how activists and non-activists perceive and evaluate each other. This is important because activists often depend on societal support to achieve their goals. We examined these perceptions and evaluations in three field experiments set in different contexts, i.e., student protests in the Netherlands 2014/2015 (Study 1, activist sample $N$ = 190; Study 2, non-activist sample $N$ = 145), and environmental protests in Paris in 2015 (Study 3, activist sample $N$ = 112). Through a scenario method, we manipulated the motivations expressed for (in)action by a member of the other group (i.e., an activist or non-activist) and measured individuals' perceptions and evaluations. Findings showed that activists perceived a non-activist as selfish and felt personally distant to them, especially when a non-activist dismissed moral obligation for action (Study 1 and 3). By contrast, non-activists had a rather positive view of activists, especially in response to an activist communicating collective concerns for action (Study 2). Study 4 ($N$ = 103) further supported this pattern of findings by showing that activists perceived larger intergroup differences than non-activists. We conclude that mutual perceptions and evaluations of activists and non-activists are asymmetrical, which may have (negative) consequences for mobilization for social change.

## Introduction

In order to maximize chances of achieving social change, activists need to mobilize their fellow disadvantaged group members and gain broader societal support for their cause. Indeed, the political struggle between those who seek social change and those who oppose it, often entails a competition over support from the broader society [1, 2]. Against this backdrop, one of the main goals of demonstrations and protests is to reach out to and win the "hearts and minds" of those within the broader ingroup, as well as third parties [3, 4], because the potential success of social movements depends on a strong and positive bond between activists and their potential followers.

However, the literature suggests that activists and non-activists differ in their concerns and motivations. Activists are often seen as 'idealists', who are highly committed to the group's cause; by contrast, the non-activists may be sympathetic to the activists' cause, but do not engage in collective action [5]. This implies that activists may not share a common ground

**Competing interests:** The authors have declared that no competing interests exist.

with non-activists, which may even make them reluctant to rely on non-activists to achieve their goals.

Little is known however, about how activists and non-activists perceive each other—as friends or foes. Importantly, the few pointers available in the literature hint in different directions. Activists sometimes see non-activists as potential allies and resources to be mobilized [6]. At the same time, they may also perceive non-activists as those who do not demonstrate sufficient solidarity with the group [7]. From the point of view of non-activists, some work suggests that activists may be perceived as 'heroes' or, quite differently, as 'complainers' [8, 9]. It is of both theoretical and practical relevance to examine how these groups perceive and evaluate each other—after all, negative relations between these groups may stand in the way of the change activists want to see in the world. We therefore conducted four studies that aimed to map the potentially different perceptions that members of these groups may have of each other.

## Defining activists and non-activists

Collective action is commonly defined as any action undertaken by individuals on behalf of a group to achieve group goals [10]. The social-psychological literature on collective action identifies four key motivations to engage in collective action: individuals' psychological ties with the group (i.e., group identification), perceptions of group-based injustice and the resulting feelings of anger, perceived violations of individual and/or group-based moral beliefs, and group efficacy beliefs [10, 11]. We use participation in collective action and membership in a social movement as the criterion to define activists and non-activists. Accordingly, activists are individuals who engage in various forms of collective action, such as protests, demonstrations, building occupation, etc. Moreover, some activists are formal members of political movements and they may also be involved in organizing and mobilizing others to take part in them [12, 3]. We thus consider both inexperienced, first-time protesters and long term, committed members of social movements as activists. Our definition circumvents the problem with the 'activist' label, because some activists may not necessarily endorse it prior to participation in collective action [13], or may even explicitly reject it [14].

Similarly, by defining non-activists as those who do not participate in collective action and/or are not members of political movements, we include those who may belong to the same social group as activists (e.g., ethnic, gender, religious group), as well as to third parties and/or general public. In the literature on political solidarity, they are referred to as the silent majority [2]. Importantly, we do not assume that non-activists are not necessarily uninvolved or passive, as they may still engage in individual actions that align with the activists' cause (e.g., individual pro-environmental behavior), but they do not take part in collective action.

Moreover, prior research suggests that activists and non-activists have different motivational profiles. For instance, activists feel stronger psychological ties with their group and more so with their activist subgroup [15, 10]. In addition, they often feel morally obliged to address injustice [16, 17]. In contrast to activists, non-activists are more concerned with individual rather than collective goals, and focused on "rational" calculations of individual benefits and costs of participation [18, 6], than on moral reasons to act [19]. In general, activists are more likely to be driven by collective and moral motivations [16, 19, 11], whereas non-activists are more likely to be driven by individual and instrumental concerns [18, 5]. We opted against a motivational definition of activism, because activists' and non-activists' motivations may vary greatly depending on the context of collective action [20]. Additionally, our definition allows for the possibility that both activist and non-activists may communicate different motivations for their behavior, which may affect how they perceive and evaluate each other.

## Mutual perceptions and communication of motivations for action

Examining mutual perceptions between activists and non-activists is an important question, because one of the key goals of collective action is to mobilize public support for the activists' cause [3]. Furthermore, positive relations between activists and their potential followers are perceived to be the key to a movement's success [21, 1, 2]. However, the existing work suggests that activists and non-activists may not always see eye to eye. For example, research on confrontation of discrimination suggests that those who address injustice are appreciated and receive support, at least from their own group [22]. Nonetheless, more often the confronters run the risk of being perceived as 'complainers', with their actions being evaluated as inappropriate [23, 8]. Moreover, members of the general public endorsing negative stereotypes of activists (e.g., perceive them as militant or eccentric) are less likely to adopt the behaviors they promote [24]. At the same time, activists may judge harshly and/or disidentify from their broader ingroup, if they perceive them as lacking commitment to the group cause [25, 26], or failing to show solidarity [27, 7]. This suggests that activists may perceive non-activists as too selfish (i.e., focusing too much on their individual interests, as opposed to group interests), whereas non-activists may see activists as complainers.

We propose that how activists and non-activists perceive each other depends not only on what they do (or not do), but also on the reasons they communicate for their behavior. Below we elaborate on the hypotheses for activists and non-activists separately.

First, non-activists may explain their decision not to participate by denying that the protest will have any effect on the power holders (i.e., instrumental motivation), or by denying the moral obligation to fight for social change (i.e., moral motivation). Previous research found that highly politicized individuals (e.g., feminists) identify less with other members of their group (e.g., women), if they do not moralize the activists' cause [26], because denying moral basis for action goes directly against what activists believe in [16, 17, 19]. Faith in the effectiveness of collective action in affecting social change facilitates participation, however, even activists do not always believe in it [3]. Thus, we expect that activists will have a more negative view of a non-activist who communicates moral rather than instrumental reasons for inaction (*Hypothesis 1*).

Moreover, non-activists can justify lack of participation by communicating individual or collective reasons for inaction. Concretely, they may emphasize that their individual vs. collective presence will not have any effect or that they do not feel personally vs. collectively morally obliged to act. Framing inaction as a collective decision rather than an individual decision, should be judged more negatively by activists (*Hypothesis 2*), because it threatens more strongly the beliefs of activists who see themselves as acting for and on behalf of the broader group [5]. However, we also acknowledge the possibility that both framings may not be received well at all by activists. Indeed, in the context of pro-environmental behavior, for example, some environmental activist groups believe individual engagement and responsibility are essential in achieving social change [28, 29]. As a consequence, it is possible that individualized framing of inaction could be seen as equally opposed to activists' beliefs as a collective framing of inaction.

When it comes to non-activists' views of activists, we propose that non-activists will have a more positive view of an activist who communicates moral motivation for collective action in comparison to an activist who communicates instrumental motivation for collective action (*Hypothesis 3*). For example, prior work on confrontation of discrimination suggests that confronters who act out of self-interest are more likely to be perceived as complainers by perceivers than those who act out of altruistic or moral reasons [30]. Moreover, sociological research on mobilization and framing suggests that moral communication from social movements is

more persuasive and successful [31]. Consequently, communicating moral rather than instrumental motivation for action should match better the expectations of who the activists are and lead to evaluations that are more positive.

Moreover, activists who emphasize collective rather than individual reasons (irrespective of whether they are instrumental or moral) for participation may be liked more by non-activists (*Hypothesis 4*). This is because activists are "entrepreneurs of identity" [32, 33], whose task is to define the audience as part of the common identity. Communicating shared identities signals inclusion, and can help activists define the non-activists as part of their group [21, 4, 1], which in turn should make them more popular among non-activists. Finally, prior literature on the negative relations between strikers and strike-breakers [27, 7], suggests that activists may be more likely to glorify their group and conversely disparage those who fail to act. Thus, we expect activists to perceive greater differences between themselves and the non-activists, by evaluating their own group more positively and the non-activists more negatively, whereas we expect non-activists to feel equally positive about both groups (*Hypothesis 5*).

## Overview of studies

To test these hypotheses, we conducted a set of three field experiments with activists (Study 1 and Study 3) and non-activists (Study 2) that allowed us to experimentally vary the communication of moral vs. instrumental and collective vs. individual reasons for (in)action as expressed by the other group. In all studies, we asked participants to evaluate a fictitious member of the other group on three dimensions: sociability, morality and rationality. Moreover, we examined the psychological distance between activists and non-activists by asking the participants how personally close they feel to the member of the other group and whether they share the same group membership. Study 1 and Study 3 explored activists' perceptions of non-activists and provided experimental tests for *Hypothesis 1* and *Hypothesis 2*. Importantly, we emphasized that a non-activist supported the activists' cause, but decided not to act for various reasons. This was done to eliminate the possibility that the effects are driven by the perceived differences in opinions, which was already shown by prior work [26], rather than motivations for inaction. Study 2 was conducted with non-activists, which enabled us to test *Hypothesis 3* and *Hypothesis 4*. In Study 4, we approached both groups at the same time (i.e., activists and general public). We did not manipulate the motivations for (in)actions, but we examined the support for *Hypothesis 5* by investigating how activists and non-activists evaluate both subgroups at the same time.

## Study 1

### Method

**Participants and procedure.** The sample consisted of 190 students attending the demonstration against government measures in November, 2014 in The Hague (see http://www.omroepwest.nl/nieuws/2715375/Studentenprotest-op-Malieveld-Den-Haag-4-arrestaties-en-Jet-Bussemaker-bekogeld-met-tomaten). 90% of the data was collected during the four-hour protest in The Hague; relatively small number of questionnaires was collected during the train ride with one of the student activist groups to The Hague. The response rate was high: 77% of the total number of protesters approached (i.e., 248) agreed to take part in the study. We did not determine a priori the sample size, but focused on getting as many participants as possible. The same participants filled out a second questionnaire that was created by another team of researchers from the University of Groningen. This questionnaire asked about activists' emotions. We were not involved in creation of this questionnaire, and to our knowledge this work has not been published yet. The study was approved by the ethics committee of University of

Groningen. All participants were given a written informed consent, which they signed before they took part in the study.

Three participants were excluded from the analyses: two turned out to be younger than 16 years and could not participate in a study without a parental consent according to the Dutch law; one person failed to fill out the survey completely. The sample was relatively of young age ($M_{age}$ = 19.96; $SD$ = 4.68, range 16–46; 51.9% women, 41.7% men, 6.4% did not disclose). The sample consisted mostly of less experienced first-time protesters (only 26.7% were members of a politically engaged student organization).

**Materials.** *Manipulation.* The participants were randomly assigned to one of the four conditions and were asked to read about a fellow student *J.* who did not come to the demonstration. The fictitious fellow student *J.* explained their decision not to come to the protest by referring to either moral or instrumental reason. Moreover, we manipulated whether the reason was communicated as an individual or a collective decision (collective framing is provided in the brackets). More specifically, in the two moral conditions participants read about *J.* who sympathized with the cause, but said ". . . I believe it's not my personal moral obligation [our collective moral obligation as current students] to fight for an equal access to education in the future [for future students]". In the two instrumental conditions, *J.* said ". . .I believe that my personal presence [our collective presence at the protest as current students] will not have an effect on the government's plans for education in the future [for future students]".

*Character evaluation.* We asked the participants to judge the fellow activist student on perceived sociability (i.e., adjectives selfish, egoistic, arrogant and social), morality (honest, moral and principled) and rationality (irrational, realistic and pragmatic). The term pragmatic was excluded from analyses, because majority of our participants did not seem to know what it meant. We used a seven point Likert scale (1- *Not at all* to 7 –*Very much*) for all items. The principal component analyses with Oblimin rotation extracted three weakly correlated factors (correlations ranged from |.14| to |.20|) reflecting the three dimensions, with eigenvalues larger than one, which explained 60.37% variance. The item social loaded on the moral dimension. We used the original items to calculate an average rating for perceived selfishness (items: selfish, egoistic, and arrogant; Cronbach's α = .77), perceived morality (items: social, honest, moral, and principled; Cronbach's α = .65) and perceived rationality of the non-activist (items: irrational [reverse coded] and realistic, $r[180]$ = .26, $p < .001$).

*Psychological distance to the non-activist.* First, the participants rated how representative the non-activist is of the larger ingroup (i.e., all students), the activist group (i.e., the group opposing the government's measures), and the adversary group (i.e., the group supporting the government's measures). Second, we asked the participants to what extent they felt personally close to *J.*

*Demographics.* At the end of the survey, participants filled out the demographics: age, gender, education and political orientation (1-*Left* to 7-*Right*). The participants identified on average as moderate, but leaning to the left of the political spectrum ($M$ = 3.36, $SD$ = 1.67). We also had additional questions about participants' perceptions of the issue and identification with the group (for more details please see Supplementary Materials), their motivations for joining (vs. not joining the protest), and their feelings towards the non-activist. The complete questionnaires and datasets for each study can be found here: https://osf.io/72yjc/.

## Results and discussion

**Character evaluation.** We ran a two-way Multivariate Analysis of Variance (MANOVA) with Motivation (Moral vs. Instrumental) and Framing (Individual vs. Collective) as between-subject factors with our three dependent variables. The analysis yielded a significant

multivariate main effect of Motivation, Wilks' Lambda = .95, $F(3,180) = 3.02$, $p = .031$, $\eta p^2 =$ .05; but no significant effect of Framing, Wilks' Lambda = .98, $F(3,180) = 1.14$, $p = .336$, $\eta p^2 =$ .02, or interaction, Wilks' Lambda = .98, $F(3,180) = 1.13$, $p = .337$, $\eta p^2 = .02$. Follow- up univariate analyses revealed only one significant main effect of Motivation on selfishness ratings, $F(1,188) = 8.74$, $p = .004$, $\eta p^2 = .05$. In line with *Hypothesis 1*, non-activist was perceived as more selfish in moral in contrast to instrumental conditions ($M_{\text{moral}} = 4.24$, $SE = 0.14$ vs. $M_{\text{instrumental}} = 3.65$, $SE = 0.14$). There were no other effects. Overall, activists did not perceive the non-activist as particularly rational ($M = 3.99$, $SD = 1.24$) or moral ($M = 3.75$, $SD = 1.11$). Means, standard deviations and univariate analyses are reported in Table 1.

**Psychological distance to the non-activist.** Importantly, the activists thought that the non-activist was not a representative of the larger student group ($M = 3.19$, $SD = 1.56$), nor the activist group ($M = 2.23$, $SD = 1.52$). Rather, they saw the non-activist as someone representing the adversary group ($M = 3.45$, $SD = 1.78$). However, there were no effects of manipulations on these variables: Motivation, Wilks' Lambda = .98, $F(3,176) = 1.43$, $p = .232$, $\eta p^2 = .02$; Framing, Wilks' Lambda > .99, $F(3,176) = 0.24$, $p = .869$, $\eta p^2 < .01$, or interaction, Wilks' Lambda = .99, $F(3,176) = 0.39$, $p = .762$, $\eta p^2 = .01$. There was only one significant effect of Motivation on personal closeness, $F(1,181) = 3.95$, $p = .048$, $\eta p^2 = .02$. The activists perceived themselves as more personally distant to the non-activist who communicated moral reason for inaction ($M = 2.46$, $SE = 0.15$) than to the non-activist who communicated instrumental reason for inaction ($M = 2.90$, $SE = 0.16$).

In sum, in line with previous research [25, 26], those who went to the protest did not think too highly of those who stayed at home. We found some, but weak, support for *Hypothesis 1* such that using moral justification for inaction increased perceptions of selfishness and further decreased feelings of closeness. On the other hand, we did not find support for *Hypothesis 2*, because communicating the decision as an individual or a collective one did not seem to matter to activists. In Study 2, we turned to non-activists and investigated their views of those who

**Table 1. Study 1: Student activists' evaluation of a non-activist.**

| Dependent variables | Moral motivation | | Instrumental motivation | | Motivation | Framing | Motivation * Framing |
|---|---|---|---|---|---|---|---|
| | Individual | Collective | Individual | Collective | | | |
| | (n = 50) | (n = 48) | (n = 44) | (n = 45) | | | |
| *Character evaluation*: | | | | | | | |
| Selfish | 4.03 (1.32) | 4.45 (1.46) | 3.67 (1.32) | 3.63 (1.30) | $F(1,182) = 8.74$, $p = .004$, $\eta_p^2 = .05$ | $F(1, 182) = 0.91$, $p = .341$, $\eta_p^2 = .005$ | $F(1, 182) = 1.40$, $p = .238$, $\eta_p^2 = .01$ |
| Moral | 3.76 (1.20) | 3.68 (1.32) | 3.77 (0.87) | 3.81 (1.01) | $F(1,182) = 0.15$, $p = .698$, $\eta_p^2 = .001$ | $F(1,182) = 0.01$, $p = .909$, $\eta_p^2 < .001$ | $F(1,182) = 0.13$, $p = .722$, $\eta_p^2 = .001$ |
| Rational | 3.94 (1.30) | 3.89 (1.23) | 3.82 (1.36) | 4.35 (1.00) | $F(1,182) = 0.91$, $p = .341$, $\eta_p^2 = .005$ | $F(1,182) = 1.76$, $p = .186$, $\eta_p^2 = .01$ | $F(1,182) = 2.65$, $p = .105$, $\eta_p^2 = .01$ |
| *Psychological distance to the non-activist*: | | | | | | | |
| Represents all students | 3.08 (1.48) | 3.04 (1.67) | 3.51 (1.61) | 3.16 (1.46) | $F(1,178) = 1.38$, $p = .241$, $\eta_p^2 = .01$ | $F(1,178) = 0.72$, $p = .397$, $\eta_p^2 = .004$ | $F(1,178) = 0.45$, $p = .502$, $\eta_p^2 = .003$ |
| Represents activist group | 2.04 (1.09) | 2.13 (1.64) | 2.56 (1.74) | 2.25 (1.57) | $F(1,178) = 2.27$, $p = .134$, $\eta_p^2 = .01$ | $F(1,178) = 0.16$, $p = .690$, $\eta_p^2 = .001$ | $F(1,178) = 0.93$, $p = .336$, $\eta_p^2 = .005$ |
| Represents adversary group | 3.65 (1.69) | 3.55 (2.00) | 3.26 (1.68) | 3.32 (1.78) | $F(1,178) = 1.38$, $p = .242$, $\eta_p^2 = .01$ | $F(1,178) = 0.03$, $p = .955$, $\eta_p^2 < .001$ | $F(1,178) = 0.09$, $p = .771$, $\eta_p^2 < .001$ |
| Individual closeness | 2.51 (1.53) | 2.40 (1.48) | 2.93 (1.53) | 2.87 (1.50) | $F(1,181) = 3.95$, $p = .048$, $\eta_p^2 = .02$ | $F(1,181) = 0.15$, $p = .701$, $\eta_p^2 = .001$ | $F(1,181) = 0.01$, $p = .927$, $\eta_p^2 < .001$ |

Unadjusted means (and standard deviations).

went to the protest. Overall, we expected non-activists to have less positive views of activists who use individual or instrumental motivations to justify participation in the protest.

## Study 2

### Method

**Participants, procedure and design.** The sample consisted of 145 students enrolled at the University of Groningen in the Netherlands. Two research assistants approached students in the canteens and libraries during last two weeks in January 2015. The average age of this sample was relatively young ($M_{age}$ = 20.2; $SD$ = 2.09, range 17–29; 65% women, 34.3% men, one person did not disclose their gender) and the participants were university students from various disciplines. The participants were not politically active and we excluded two students who were members of a political organization (one of them also went to the protest). Therefore, they fit with our definition of non-activists and differ from the sample of activists in Study 1. The study took about 5 minutes. The study was approved by the ethics committee of University of Groningen. All participants were given a written informed consent, which they signed before they took part in the study.

**Materials.** *Manipulation*. We adjusted the manipulations to reflect the motivation of an activist. The participants were randomly assigned to one of the four conditions and they read about a fictitious fellow student *J*. who went to the demonstration, because they believed ". . . it's my personal moral obligation [our collective moral obligation as current students] to fight for an equal access to education in the future [for future students]". In the two instrumental conditions, *J*. said ". . .I believe that my personal presence [our collective presence at the protest as current students] will have an effect on the government's plans for education in the future [for future students]".

**Dependent variables.** *Character evaluation*. We asked the same items as in Study 1. The principal component analysis with Oblimin rotation extracted two factors with eigenvalues larger than one, which explained 54.67% variance. Items pertaining to sociability and rationality loaded on the first factor, whereas the morality items (including item social) loaded on the second factor (the factors were moderately correlated, $r$ = .38). We used the original items to calculate an average rating for selfish (items: selfish, egoistic, and arrogant; Cronbach's $\alpha$ = .77) and moral scale (items: social, honest, moral, and principled; Cronbach's $\alpha$ = .69). We decided to keep the rational scale as separate (items: irrational [reverse coded] and realistic, $r$ [141] = .40, $p$ < .001).

*Psychological distance to the activist*. We asked how representative was the activist of the broader ingroup (i.e., all students), the activist group and how personally close our participants felt to *J*.

*Demographics*. At the end of the survey, participants filled out the demographics and political orientation (1-*Left* to 7-*Right*). The sample was on average of moderate political orientation ($M$ = 3.81, $SD$ = 1.26).

### Results and discussion

**Character evaluation.** A two-way MANOVA with Motivation (Moral vs. Instrumental) and Framing (Individual vs. Collective) as between- subject factors yielded a significant multivariate main effect of Motivation, Wilks' Lambda = .93, $F(3,137)$ = 3.59, $p$ = .015, $\eta p^2$ = .07; again no significant effect of Framing, Wilks' Lambda = .97, $F(3,137)$ = 1.55, $p$ = .205, $\eta p^2$ = .03, or interaction, Wilks' Lambda = .99, $F(3,137)$ = 0.67, $p$ = .573, $\eta p^2$ = .01. Follow-up univariate analyses revealed an effect of Motivation on perceptions of rationality $F(1,139)$ = 7.42, $p$ = .007, $\eta p^2$ = .05, and an effect of Framing on perceptions of selfishness $F(1,139)$ = 4.02, $p$ = .047,

**Table 2. Study 2: Student non-activists' evaluation of an activist.**

| Dependent variables | Moral motivation | | Instrumental motivation | | Motivation | Framing | Motivation * Framing |
|---|---|---|---|---|---|---|---|
| | Individual | Collective | Individual | Collective | | | |
| | (n = 36) | (n = 35) | (n = 38) | (n = 34) | | | |
| *Character evaluation*: | | | | | | | |
| Selfish | 2.56 (1.04) | 2.46 (1.05) | 2.76 (0.87) | 2.25 (0.69) | $F(1,139) = 0.001$, $p = .977$, $\eta_p^2 < .001$ | $F(1,139) = 4.02$, $p = .047$, $\eta_p^2 = .03$ | $F(1,139) = 1.80$, $p = .182$, $\eta_p^2 = .01$ |
| Moral | 5.17 (0.74) | 5.25 (0.67) | 4.98 (0.77) | 5.08 (0.68) | $F(1,139) = 2.23$, $p = .137$, $\eta_p^2 = .02$ | $F(1,139) = 0.55$, $p = .458$, $\eta_p^2 = .004$ | $F(1,139) = 0.01$, $p = .928$, $\eta_p^2 < .001$ |
| Rational | 4.56 (1.22) | 4.74 (1.00) | 3.95 (1.05) | 4.37 (1.02) | $F(1,139) = 7.42$, $p = .007$, $\eta_p^2 = .05$ | $F(1,139) = 2.83$, $p = .095$, $\eta_p^2 = .02$ | $F(1,139) = 0.42$, $p = .52$, $\eta_p^2 = .003$ |
| *Psychological distance to the activist*: | | | | | | | |
| Represents all students | 4.19 (1.45) | 4.89 (1.08) | 4.22 (1.18) | 4.56 (1.05) | $F(1,139) = 0.57$, $p = .452$, $\eta_p^2 = .004$ | $F(1,139) = 6.55$, $p = .012$, $\eta_p^2 = .045$ | $F(1,139) = 0.74$, $p = .39$, $\eta_p^2 = .005$ |
| Represents activist group | 5.53 (1.08) | 5.57 (1.42) | 5.43 (1.04) | 5.35 (1.30) | $F(1,138) = 0.59$, $p = .444$, $\eta_p^2 = .004$ | $F(1,138) = 0.01$, $p = .93$, $\eta_p^2 < .001$ | $F(1,138) = 0.09$, $p = .764$, $\eta_p^2 = .001$ |
| Individual closeness | 3.83 (1.63) | 4.34 (1.33) | 3.53 (1.45) | 3.97 (1.27) | $F(1,139) = 2.02$, $p = .157$, $\eta_p^2 = .01$ | $F(1,139) = 3.98$, $p = .048$, $\eta_p^2 = .03$ | $F(1,139) = 0.02$, $p = .892$, $\eta_p^2 < .001$ |

Unadjusted means (and standard deviations).

$\eta_p^2 = .03$. Specifically, *J.* was perceived as more rational in moral in contrast to instrumental conditions ($M_{\text{moral}} = 4.65$, $SE = 0.13$ vs. $M_{\text{instrumental}} = 4.16$, $SE = 0.13$). Moreover, in line with *Hypothesis 4*, when decision to participate in the protest was framed as a collective rather than as a personal decision, the activist was perceived as even more selfless ($M_{\text{collective}} = 2.35$, $SE = 0.11$ vs. $M_{\text{individual}} = 2.66$, $SE = 0.11$). Means, standard deviations and univariate analyses are reported in Table 2.

**Psychological distance to the activist.** A two-way MANOVA on perceptions of representativeness yielded a significant multivariate main effect of Framing, Wilks' Lambda = .95, $F(2,137) = 3.70$, $p = .027$, $\eta_p^2 = .05$; but no significant effect of Motivation, Wilks' Lambda = .99, $F(2,137) = 0.44$, $p = .647$, $\eta_p^2 = .01$, or interaction, Wilks' Lambda > .99, $F(2,137) = 0.37$, $p = .692$, $\eta_p^2 = .01$. Follow-up univariate analyses revealed that the activist was judged as a more representative member of the broader ingroup (i.e., all students) in the collective conditions than in individual conditions ($M_{\text{collective}} = 4.72$, $SE = 0.15$ vs. $M_{\text{individual}} = 4.21$, $SE = 0.14$), $F(1,138) = 6.55$, $p = .012$, $\eta_p^2 = .05$. Furthermore, the non-activists perceived themselves as somewhat more personally close to the activist when the communication emphasized the collective as opposed to individual motives for attending the protest ($M_{\text{collective}} = 4.16$, $SE = 0.17$ vs. $M_{\text{individual}} = 3.68$, $SE = 0.17$), $F(1,139) = 3.98$, $p = .048$, $\eta_p^2 = .03$.

In sum, in Study 2 we did not find evidence that the activist was perceived as a complainer [22, 8]. On the contrary, the findings indicated, as expected, that the non-activists thought rather positively of those who went to the demonstration. We found some support for *Hypothesis 3*, because our participants perceived an activist who expressed moral obligation in contrast to instrumental motivation as more realistic. Overall however, expressing moral or instrumental motivation had very little effect on non-activists' perceptions of activists. Importantly, and in support of *Hypothesis 4*, when an activist communicated collective motivations, they were seen as more psychologically close (both on an individual level, and as a more representative group member) to those who did not participate.

Interestingly, the findings of Study 1 and 2 suggest (perhaps unjustly) that activists have somewhat more negative views of non-activists than vice versa. Thus, in Study 3 we decided to

focus again on activists, because movement building depends primarily on the activists reaching out to their broader ingroup or the general public for their support [1, 2]. Moreover, the findings of Study 1 may be limited due to our choice of context, participants' characteristics and stimuli. We tried to address all these concerns in Study 3.

## Study 3

We made several changes in Study 3. First, in Study 1, activists and non-activists belonged to the same group and presumably shared the same norms. It is possible that activists held a bitter grudge against those who did not show up, because their absence directly undermines the unity of the group in the face of adversaries, like in the case of strikers and strikebreakers [27, 7]. Perhaps, in the context of issue-based activism, like environmentalism [12], where there is no available group identity with clearly defined group norms about what group members should do or not do, activists may not be so harsh towards those who are not there. Furthermore, the reason why we did not find many effects of our manipulations in Study 1 may be due to our sample, which consisted of students without much experience with activism. Prior research finds that more 'seasoned' activists place higher importance on movement building than novices who are more concerned with influencing power holders [34, 3]. Hence, more experienced activists may be more understanding of those who do not take part in action than less experienced activists. To address these two issues, we changed the context to a pro-environmental protest in Study 3, and we reached out to a more experienced sample of activists.

Moreover, activists' relatively negative views of non-activists in Study 1 may be due to our choice of motivations for inaction that emphasize concerns that directly violate activists' beliefs. However, some non-activists may not show up at a protest, but still contribute to the collective goal by engaging in a range of individual behaviors, such as boycotting classes to support the student movement, or buying ecological products in case of pro-environmental movement [35, 36]. Hence, in Study 3 we included an additional justification for inaction, which reflects the preference for engagement in individual action over collective action as a mean to achieve the collective goal. This motivation should resonate better with the activist group, because it emphasizes taking personal responsibility to achieve social change. Thus, we expected that the activist would evaluate this subgroup more positively and especially in comparison to those who deny moral obligation to act.

### Method

**Participants and procedure.** Five research assistants approached people who took part in demonstrations and sittings in Amsterdam and Paris during United Nations Climate Change Talks in Paris in 2015 (COP 21). The participants were asked to participate in a study on environmental activism, and those who agreed received a link via email to the study, which was administered via Qualtrics. Thus, in contrast to Study 2, the participants filled out the questionnaire after the events were over. Due to the international character of this event, the study was translated to English, Dutch and French. The study was approved by the ethics committee of University of Groningen.

From the 710 email addresses we collected 162 individuals responded. The experiment was administered at the very end of a larger survey on environmental activism, reducing the final number of participants who took part in the experiment to 112. The broader survey included the Schwartz value questionnaire and various questionnaires on environmental behavior and identity, and this part of the data has been published [37]. This resulted in 50 participants dropping out, though well before the experiment was introduced, which suggests that the non-response may have selected for the most motivated participants.

76 participants answered the demographics questions ($M_{age}$ = 30.19; $SD$ = 9.08; range 20–67; 55.3% women, 36.8% men, 7.9% other). In contrast to Study 1, the sample was much more diverse and consisted of more experienced activists: the participants came from 10 different countries and majority of them (i.e., 73.7%) reported being an active member of an environmental organization for on average 4 years.

**Materials.** *Manipulation.* As in Study 1, the participants read about a person *J.* who cares about the environment, but still decided not to join the demonstrations because they did not feel (individually or collectively) morally obliged to fight against climate change vs. did not believe that (individual or collective) presence at the protest will have any effect on the political powers. In the two additional conditions, *J.* expressed that . . . ' I, as an individual [we, as people who care about the environment] would [should] rather focus my[our] energy on 'being the change' (e.g. grow my own food, not waste . . .) instead of trying to influence international and/or national politics'. Thus, the design was Motivation (Moral vs. Instrumental vs. Personal Responsibility) x Framing (Individual vs. Collective) between-subjects design.

*Character evaluation.* We used shortened version of the scales in previous studies to evaluate the character of the non-activist: selfish (selfish and egoistic, $r$ [109] = .71, $p$ < .001), moral (moral, honest, social, Cronbach α = .61) and rational (realistic and irrational [reverse coded], $r$[109] = .60, $p$ < .001).

*Psychological distance to the non-activist.* We again asked to what extent the activists felt personally close the non-activist. Furthermore, as in Study 1, we asked to what extent the non-activist was representative of the activist group (i.e., those who care about the environment), and the adversary group (i.e., those who do not care about the environment). In addition to the broader activist group, we also specifically checked whether the non-activist was perceived as representative of those who gathered to protest the negotiations. Moreover, in order to validate our assumptions that given motivations for inaction are typically associated with non-activists, we asked our participants to evaluate the extent to which our fictitious character was representative of the general public. Lastly, we asked participants to judge whether the non-activist was an asset to the environmental movement (1 –*Detrimental to the movement* to 7-*Beneficial to the movement*).

*Demographics.* At the end of the survey, participants filled out the demographics and political orientation. The participants identified on average as leftists ($M$ = 1.78; $SD$ = 0.90).

## Results and discussion

**Character evaluation.** A two-way MANOVA with Motivation (Moral vs. Instrumental vs. Personal responsibility) and Framing (Individual vs. Collective) as between- subject factors yielded two significant multivariate main effects of Motivation, Wilks' Lambda = .78, $F(6,204)$ = 4.42, $p$ < .001, $\eta p^2$ = .12; and Framing, Wilks' Lambda = .89, $F(3,102)$ = 4.41, $p$ = .006, $\eta p^2$ = .12, but no significant interaction, Wilks' Lambda = .98, $F(6,204)$ = 0.30, $p$ = .936, $\eta p^2$ = .01. Replicating the findings from Study 1, follow-up univariate analyses revealed that the communicated motivations for inaction had an effect on perceptions of selfishness $F(2,104)$ = 7.93, $p$ = .001, $\eta p^2$ = .13, but also on the perceptions of morality of the non-activist, $F(2,104)$ = 4.21, $p$ = .017, $\eta p^2$ = .08. Using moral motivations to justify inaction was perceived as more selfish ($M$ = 4.53, $SE$ = 0.26) than using instrumental motivation ($M$ = 3.74, $SE$ = 0.25), $p$ = .03, 95% CI [0.08, 1.51], or personal responsibility ($M$ = 3.12, $SE$ = 0.24), $p$ < .001, 95%CI [0.71, 2.12]. The latter two did not differ significantly, $p$ = .076, 95%CI [-0.07, 1.30]. Denying moral obligation was also perceived as the least moral ($M$ = 3.88, $SE$ = 0.19), followed by instrumental motivation ($M$ = 4.16, $SE$ = 0.18), and personal responsibility ($M$ = 4.61, $SE$ = 0.17). Only moral obligation and individual responsibility differed significantly $p$ = .005, 95%CI [-1.23, -0.22].

**Table 3. Study 3: Environmental activists' evaluation of a non-activist: Means and standard deviations.**

| Dependent variables | Moral motivation | | Instrumental motivation | | Personal responsibility | |
|---|---|---|---|---|---|---|
| | Individual | Collective | Individual | Collective | Individual | Collective |
| | (n = 17) | (n = 17) | (n = 19) | (n = 18) | (n = 20) | (n = 21) |
| *Character evaluation*: | | | | | | |
| Selfish | 4.91 (1.43) | 4.16 (0.81) | 3.97 (1.84) | 3.50 (1.54) | 3.50 (1.68) | 2.74 (1.45) |
| Moral | 4.04 (1.03) | 3.73 (0.98) | 4.11 (1.13) | 4.22 (0.85) | 4.60 (1.15) | 4.62 (1.24) |
| Rational | 3.88 (1.84) | 3.22 (1.45) | 4.53 (1.59) | 4.28 (1.50) | 4.08 (1.54) | 3.81 (1.35) |
| *Psychological distance to the non-activist*: | | | | | | |
| . . .represents those who care for the environment | 3.29 (1.36) | 3.00 (1.28) | 4.11 (1.53) | 3.94 (1.35) | 4.80 (1.32) | 4.81 (1.50) |
| . . . represents those who do not care for the environment | 5.35 (1.62) | 4.76 (1.99) | 4.17 (2.15) | 3.94 (1.59) | 2.40 (1.76) | 3.33 (1.96) |
| . . .COP protesters | 1.59 (0.80) | 2.06 (1.64) | 2.11 (1.13) | 2.56 (1.62) | 2.40 (1.57) | 2.48 (1.29) |
| . . .general public | 5.00 (1.51) | 4.29 (1.86) | 5.17 (1.51) | 4.67 (1.28) | 4.05 (1.79) | 3.76 (1.48) |
| Individual closeness | 2.41 (1.62) | 2.18 (1.33) | 2.67 (1.61) | 2.61 (1.46) | 2.80 (1.74) | 3.10 (1.38) |
| Hindrance-Benefit | 3.00 (1.55) | 3.29 (1.86) | 2.67 (1.37) | 3.22 (1.22) | 4.30 (1.56) | 4.43 (1.75) |

Additionally, and in contrast to *Hypothesis 2*, individual framing was perceived as somewhat more selfish than the collective framing ($M_{individual}$ = 4.13, $SE$ = 0.20; $M_{collective}$ = 3.47, $SE$ = 0.21), $F(1,104)$ = 5.28, $p$ = .024, $\eta p^2$ = .05. Means, standard deviations and univariate analyses are reported in Tables 3 and 4.

**Psychological distance to the non-activist.** In contrast to Study 1, multivariate analysis on representativeness variables yielded a significant multivariate main effect of Motivation, Wilks' Lambda = .66, $F(8,204)$ = 5.93, $p < .001$, $\eta p^2$ = .19; but no significant effect of Framing, Wilks' Lambda = .96, $F(4,102)$ = 1.11, $p$ = .357, $\eta p^2$ = .04, or interaction, Wilks' Lambda = .96, $F(8,204)$ = 0.59, $p$ = .785, $\eta p^2$ = .02. Follow-up univariate analyses revealed a significant effect of Motivation on the extent to which the non-activist was seen as representative of the group that cares about the environment $F(2,105)$ = 13.10, $p < .001$, $\eta p^2$ = .20, the group that does not care about the environment $F(2,105)$ = 13.09, $p < .001$, $\eta p^2$ = .20, as well as the general public

**Table 4. Study 3: Environmental activists' evaluation of a non-activist: univariate analysis output.**

| Dependent variables | Motivation | Framing | Motivation * Framing |
|---|---|---|---|
| Selfish | $F(2,104)$ = 7.93, $p$ = .001, $\eta_p^2$ = .13 | $F(1, 104)$ = 5.28, $p$ = .024, $\eta_p^2$ = .05 | $F(2, 104)$ = 0.11, $p$ = .896, $\eta_p^2$ = .002 |
| Moral | $F(2,104)$ = 4.21, $p$ = .017, $\eta_p^2$ = .08 | $F(1,104)$ = 0.08, $p$ = .784, $\eta_p^2$ = .001 | $F(2,104)$ = 0.37, $p$ = .69, $\eta_p^2$ = .01 |
| Rational | $F(2, 104)$ = 2.67, $p$ = .074, $\eta_p^2$ = .05 | $F(1,104)$ = 1.78, $p$ = .186, $\eta_p^2$ = .02 | $F(2,104)$ = 0.20, $p$ = .821, $\eta_p^2$ = .004 |
| . . .represents those who care for the environment | $F(2,105)$ = 13.10, $p < .001$, $\eta_p^2$ = .20 | $F(1,105)$ = 0.32, $p$ = .573, $\eta_p^2$ = .003 | $F(2,105)$ = 0.11, $p$ = .894, $\eta_p^2$ = .002 |
| . . .represents those who do not care for the environment | $F(2, 105)$ = 13.09, $p < .001$, $\eta_p^2$ = .20 | $F(1, 105)$ = 0.13, $p$ = .908, $\eta_p^2 < .001$ | $F(2, 105)$ = 1.75, $p$ = .179, $\eta_p^2$ = .03 |
| . . .COP protesters | $F(2,106)$ = 2.04, $p$ = .135, $\eta_p^2$ = .04 | $F(1,106)$ = 1.59, $p$ = .211, $\eta_p^2$ = .015 | $F(2, 105)$ = 0.25, $p$ = .781, $\eta_p^2$ = .005 |
| . . .general public | $F(2, 105)$ = 4.32, $p$ = .016, $\eta_p^2$ = .08 | $F(1, 105)$ = 2.78, $p$ = .099, $\eta_p^2$ = .03 | $F(2, 105)$ = 0.27, $p$ = .848, $\eta_p^2$ = .003 |
| Individual closeness | $F(2,105)$ = 1.70, $p$ = .189, $\eta_p^2$ = .03 | $F(1,105) < 0.001$, $p$ = .996, $\eta_p^2 < .001$ | $F(2,105)$ = 0.30 $p$ = .745, $\eta_p^2$ = .006 |
| Hindrance-Benefit | $F(2,104)$ = 9.28, $p < .001$, $\eta_p^2$ = .15 | $F(1,104)$ = 1.18, $p$ = .28, $\eta_p^2$ = .01 | $F(2,104)$ = 0.18, $p$ = .837, $\eta_p^2$ = .003 |

Unadjusted means (and standard deviations).

$F(2,105) = 4.32$, $p = .016$, $\eta p^2 = .08$. In line with the effects on character judgments, the non-activist who communicated moral motivation for inaction was perceived as less representative of those who care about the environment ($M = 3.15$, $SE = 0.24$), than the non-activist who communicated instrumental motivation ($M = 4.03$, $SE = 0.23$), $p = .010$, 95%CI [-1.54, -0.22], or the non-activist who communicated personal responsibility ($M = 4.81$, $SE = 0.22$), $p < .001$, 95%CI [-2.30, -1.02]. The latter two also differed significantly, $p = .017$, 95%CI [-1.41, -0.15]. The non-activists who denied moral obligation to participate in collective action was seen as the most representative of those who do not care about the environment ($M = 5.06$, $SE = 0.32$), followed by the non-activist who communicated instrumental motivation ($M = 4.06$, $SE = 0.31$), $p = .026$, 95%CI [0.12, 1.88], and personal responsibility ($M = 2.87$, $SE = 0.29$), $p < .001$, 95%CI [1.34, 3.05]. The latter two also differed significantly, $p = .006$, 95%CI [0.35, 2.03].

Moreover, the non-activist who communicated moral ($M = 4.65$, $SE = 0.27$) or instrumental motivation for inaction ($M = 4.92$, $SE = 0.26$) was perceived as more representative of the general public; the difference between the two was not significant, $p = .474$, 95%CI [-1.01, 0.48]. In contrast, the non-activist who communicated personal responsibility was seen as significantly less representative of the general public ($M = 3.91$, $SE = 0.25$), than the other two: $p = .044$, 95%CI [-1.46, -0.02]; $p = .006$, 95%CI [-1.72, -0.30]. However, the non-activist was not seen as representative of those who gathered at the protest, irrespective of the motivations communicated ($M = 2.22$; $SD = 1.38$), $F(2,105) = 2.04$, $p = .135$, $\eta p^2 = .04$.

The activists did not feel personally close to those who were not there ($M = 2.65$; $SD = 1.52$), but we did not find significant effects of our manipulations on perception of closeness. Lastly, we found a significant effect of Motivation on perceptions of the non-activists as beneficial to the movement, $F(2,104) = 9.28$, $p < .001$, $\eta p^2 = .15$. The activists believed that the non-activist who used either moral ($M = 3.15$; $SE = 0.27$) or instrumental motivation ($M = 2.94$; $SE = 0.26$) as an argument for inaction was a hindrance to the environmental movement; the two did not differ significantly $p = .593$, 95%CI [-0.55, 0.95]. The participants were somewhat more positive about the non-activist who communicated personal responsibility ($M = 4.36$; $SE = 0.25$) and this condition differed significantly from the other two: $p = .001$, 95%CI [0.49, 1.95]; $p < .001$, 95%CI [0.71, 2.13].

Replicating the findings of Study 1, activists disliked the most a member of the general public who denied moral obligation to act (*Hypothesis 1*), and they believed that both those who deny moral obligation or instrumentality are a hindrance to the movement. Interestingly, they viewed those who used personal responsibility as an argument for not taking part in classical activism as more genuinely supporting the cause. Nonetheless, they did not feel personally close to the non-activist irrespective of the reasons used to justify the inaction.

Together, Study 1 and Study 3 suggest that activists do not feel very close to non-activists. However, in Studies 1–3 we only examined the perspective of each group separately, and we did not directly compare the opinions of both groups. Thus, in order to directly test *Hypothesis 5*, we conducted an additional study where we asked the participants to evaluate both groups without manipulating specific motivations for (in)action. Based on our previous findings, we expected that activists would differentiate themselves from non-activists by evaluating their own group more positively than the non-activist group. In contrast, we expected non-activists to have a generally positive view of both activists and non-activists and to differentiate less between the two groups.

## Study 4

### Method

**Context and participants.** One research assistant approached the protesters gathered in New York on June 17, 2017 to support the legislative that would outlaw nuclear weapons

(http://www.icanw.org/day-of-action/), who were mobilized by a civil group called Women's International League for Peace and Freedom. Due in part to the bad weather conditions, we managed to collect data from 22 activists present at the protest ($M_{age}$ = 34.2; $SD$ = 16.57, range 19–69; 63.6% women, 31.8% men, 4.5% other). On the same day, we collected the data from 81 Mturk workers ($M_{age}$ = 32.5; $SD$ = 10.22, range 20–65; 39.5% women, 60.5% men) living in New York, who did not participate in the protest. The study was approved by the ethics committee of University of Osnabrück. All participants were given a written informed consent, which they signed before they took part in the study.

**Materials.** *Character evaluation*. Participants were asked to evaluate both those who went to protest today and those who did not (in counterbalanced order) in terms of selfishness, morality and rationality using items used as in Study 1 and Study 2. We created two subscales that captured the extent to which activists were perceived as selfish (items: selfish, egoistic, arrogant; Cronbach's α = .87) and moral (items: social, moral, honest, principled; Cronbach's α = .85); and two subscales that captured the extent to which non-activists were perceived as selfish (items: selfish, egoistic, arrogant; Cronbach's α = .85), and moral (items: social, moral, honest, principled; Cronbach's α = .90). When the targets of judgment were activists, items irrational and realistic were significantly correlated when ($r$[101] = -.34, $p$ < .001), but not when the targets were non-activists (realistic and irrational [reverse coded], $r$[100] = -.11, $p$ = .263). We decided to run the analyses only using the item irrational.

*Perceived psychological distance*. We asked the participants to judge the extent to which both activists and non-activists were representative of those who believe nuclear weapons should be banned (i.e., the activist group) and those who do not believe that nuclear weapons should be banned (i.e., the adversary group).

*Demographics*. At the end of the survey, participants filled out the demographics and political orientation (*1*-Conservative, *7*–Progressive). The activists identified themselves as more progressive ($M$ = 5.91, $SD$ = 1.63), than non-activists ($M$ = 4.64, $SD$ = 1.78), $t$ (1,101) = - 3.02, $p$ = .003. 90.5% of the activists identified as an active member of a political organization, for more than a year, in contrast to 24.7% of non-activists. We excluded the non-activists who identified as a member of a political organization from the analyses (the analyses remained the same if included those as well, see Supplementary Materials).

## Results and discussion

**Character evaluation.** We note that due the unbalanced sample sizes, the findings of this study should be taken with caution. We ran a mixed effects ANOVA with Target Group (Target Activists vs. Target Non-activists) and Dimension (Selfishness vs. Morality vs. Irrationality) as within-subject factors and Participation (Activists vs. Non-activists) as a between-subject factor. The analysis yielded a significant effect for Dimension $F$(1.40, 121.83) = 52.68, $p$ <. 001, $\eta_p^2$ = .40, significant Target Group x Dimension interaction $F$(1.52, 121.83) = 27.40, $p$ <. 001, $\eta_p^2$ = .26, and most importantly a significant Target Group x Dimension x Participation interaction $F$(1.52, 121.83) = 14.23, $p$ <. 001, $\eta_p^2$ = .15. The main effect of Target $F$(1, 121.83) = 2.07, $p$ = .154, $\eta_p^2$ = .03, the interaction Target Group x Participation $F$(1, 121.83) = 1.52, $p$ = .221, $\eta_p^2$ = .02, and Dimension x Participation $F$(1.40, 121.83) = 1.16, $p$ = .302, $\eta_p^2$ = .01, were not significant. We used Greenhouse-Geisser corrected degrees of freedom, because the Mauchly's test indicated that sphericity could not be assumed.

In line with our expectations, activists perceived larger differences between two groups than non-activists. Concretely, they evaluated their own group as more selfless ($M_{targetactivists}$ = 2.03, $SE$ = 0.31 vs. $M_{targetnon\text{-}activists}$ = 3.44, $SE$ = 0.33), $F$(1, 80) = 17.64, $p$ <.001, $\eta_p^2$ = .18, and more moral ($M_{targetactivists}$ = 5.83, $SE$ = 0.28 vs. $M_{targetnon\text{-}activists}$ = 3.58, $SE$ = 0.29, $F$(1, 80) =

**Table 5. Study 4: Activists and general public views of both groups.**

| | Participants: | | | |
| | Non-activists | | Activists | |
| Evaluations of Targets: | M | SD | M | SD |
|---|---|---|---|---|
| Selfish activists | 2.76 | 1.46 | 2.03 | 1.35 |
| Selfish non-activists | 2.90 | 1.51 | 3.44 | 1.57 |
| Moral activists | 4.56 | 1.38 | 5.83 | 0.90 |
| Moral non-activists | 3.92 | 1.38 | 3.58 | 1.14 |
| Irrational activists | 3.02 | 1.78 | 2.62 | 1.69 |
| Irrational non-activists | 2.54 | 1.56 | 3.38 | 1.56 |
| Psychological distance: | | | | |
| . . .activists representative of people who believe that nuclear weapons should be banned | 4.97 | 2.03 | 6.30 | 1.13 |
| . . .non- activists representative of people who believe that nuclear weapons should be banned | 3.35 | 1.48 | 3.50 | 1.79 |
| . . .activists representative of people who believe that nuclear weapons should not be banned | 2.43 | 1.67 | 2.05 | 1.99 |
| . . .non- activists representative of people who believe that nuclear weapons should not be banned | 3.70 | 1.59 | 4.15 | 1.42 |

Unadjusted means (and standard deviations).

49.78, $p < .001$, $\eta_p^2 = .38$. In contrast, non-activists perceived the two groups as equally selfless ($M_{\text{targetactivists}} = 2.76$, $SE = 0.18$ vs. $M_{\text{targetnon-activists}} = 2.90$, $SE = 0.20$), $F(1, 80) = 0.52$, $p = .474$, $\eta_p^2 = .01$; and they thought the activists were more moral than non-activists: $M_{\text{targetactivists}} = 4.56$, $SE = 0.16$ vs. $M_{\text{targetnon-activists}} = 3.92$, $SE = 0.18$ $F(1, 80) = 11.82$, $p < .001$, $\eta_p^2 = .13$. There were no differences in perceptions of irrationality. Means and standard deviations are reported in Table 5.

**Perceived psychological distance.** Mixed effects ANOVA yielded a significant effect for Dimension $F(1,78) = 45.62$, $p < .001$, $\eta_p^2 = .37$, a significant, a significant Target Group x Dimension interaction $F(1,78) = 65.63$, $p < .001$, $\eta_p^2 = .46$, and a significant Target Group x Dimension x Participation interaction $F(1,78) = 4.41$, $p = .039$, $\eta_p^2 = .05$. Main effect of Target $F(1,78) = 3.06$, $p = .084$, $\eta_p^2 = .04$, Dimension x Participation interaction $F(1,78) = 2.74$, $p = .102$, $\eta_p^2 = .03$, and interaction Target x Participation $F(1,78) = 0.34$, $p = .562$, $\eta_p^2 = .004$, were not significant.

First, the activists perceived themselves as more representative of those who believe the nuclear weapon should be banned than the non-activists: $M_{\text{targetactivists}} = 6.30$, $SE = 0.42$ vs. $M_{\text{targetnon-activists}} = 3.50$, $SE = 0.37$, $F(1, 78) = 31.90$, $p < .001$, $\eta_p^2 = .29$. For non-activists, this difference was smaller: $M_{\text{targetactivists}} = 4.97$, $SE = 0.24$ vs. $M_{\text{targetnon-activists}} = 3.35$, $SE = 0.20$, $F(1, 78) = 31.90$, $p < .001$, $\eta_p^2 = .29$. Second, the activists thought that those who did not come are more representative of the adversary group: $M_{\text{targetnon-activists}} = 4.15$, $SE = 0.35$ vs. $M_{\text{targetactivists}} = 2.05$, $SE = 0.39$, $F(1, 78) = 18,72$, $p < .001$, $\eta_p^2 = .19$. Again, for non-activists, the absolute mean difference was smaller: $M_{\text{targetnon-activists}} = 3.70$, $SE = 0.20$ vs. $M_{\text{targetactivists}} = 2.43$, $SE = 0.23$, $F(1, 78) = 20.43$, $p < .001$, $\eta_p^2 = .21$. In sum, the findings indicated that activists evaluated their own fellow group members as more selfless and moral than those who did not show up. The non-activists appreciated those who went to protests, but their evaluations of both groups were more similar. Even though the findings are limited due to the small sample of activists and the unbalanced design, the differences in evaluations were consistently larger for the activist group in line with *Hypothesis 5*. One of the biggest issues with unbalanced designs is the probability that the assumption of homogeneity is violated. This may not have

necessarily been the case, because the standard deviations for both groups on all variables were similar and the tests did not indicate violation of homogeneity assumption. However, we recommend for future research to replicate the findings with a larger sample.

## General discussion

The current set of studies aimed to answer the question how activists and non-activists perceive each other, and whether their mutual perceptions are affected by the reasons they communicate for their (in)action. First, as anticipated the mutual perceptions and evaluations of activists and non-activists were opposing: activists disapproved of the non-activists (Study 1 and Study 3), whereas those who did not act positively evaluated those who did (Study 2). Second, communicating different motivations for (in)action affected how non-activists (Study 1 and Study 3) and activists were perceived by the other group (Study 2). Supporting *Hypothesis 1*, communicating moral motivation for inaction was not appreciated by activists (Study 1 and Study 3). Framing decisions for inaction in individual or collective terms did not matter so much to the activists. On the other hand, supporting *Hypothesis 4*, when the activist communicated collective reasons for action, they were perceived even more positively by non-activists (Study 2). We did not find much support for *Hypothesis 3* that communicating moral motivation for action is evaluated more positively than communicating instrumental motivation. Third, in line with *Hypothesis 5*, activists perceived larger differences between themselves and the non-activists than non-activists did (Study 4). We thus conclude that activists are more likely to distance themselves from non-activists than vice versa.

### Theoretical and practical implications

The current set of studies contributes to the literature on collective action by illuminating a blind spot, namely the mutual perceptions and evaluations of activists and non-activists. The social-psychological theorizing on social change emphasizes that third parties or general public play a crucial role in driving social change processes, and that successful social change depends on the positive relations between those who act to improve group status and the broader society [1, 2]. We identify an intriguing asymmetry between the two subgroups in terms of their perceptions and evaluations of each other.

Overall, Study 1, 3 and 4 suggest that activists, defined as those who participate in collective and/or are members of political movements, perceive non-activists in a negative light, and this seems to hold across various contexts (i.e., student protests, environmental activists and anti-nuclear activists), and applies to both mature (Study 3–4) and less experienced activists (Study 1). These negative views were especially pronounced when the non-activists, defined as those who do not participate in collective and/or are not members of political movements, negated the moral obligation to take part in the collective struggle, which goes against activists' core beliefs [16, 17, 19]. Moreover, Study 3 showed that activists believed that general public endorses selfish motivations (i.e., lack of moral commitment and denial of efficacy) to a higher extent than constructive motivation (i.e., preference to engage in individual action). Additionally, even when the reasons for inaction were not known (Study 4), or the non-activists engaged in individual action for the same cause, activists perceived a psychological distance between themselves and their audience. Negative evaluations and the exaggeration of the differences between the two groups may be the consequence of activists' need for positive distinctiveness [38]. Prior work examining activists' disidentification from the broader ingroup [25, 26] argues that it may prevent activists from securing strong ties with their audience. We echo this sentiment and suggest that activists may not fully tap into the support they may receive from non-activists.

Furthermore, the findings of Study 2 and 4 pointed out that third parties may actually be rather sympathetic toward activists. However, previous research on those who challenge discrimination warns against assuming that activists always have full support of the broader ingroup [23, 8], though the contexts described in those studies referred mostly to the cases of individual discrimination where the audience could not directly profit from the action. Importantly, the results suggest that the activists could harvest non-activists' support especially by using collective and inclusive language. If those who seek social change act as "entrepreneurs of identity" [32, 33], they can potentially pull closer the audience they are targeting, and these strategies should be particularly useful during the recruitment stage of mobilization process [6].

## Limitations

The cross-sectional design of our studies fails to capture the process of mobilization and movement building, which is more complex and dynamic than a single instance of collective action [6]. Moreover, activists and non-activists may change their views of each other after the opportunity to engage in a face-to-face interaction. For instance, both groups may gain a more nuanced understanding of each other's decisions to take vs. against taking part in collective action. Research on opinion-based groups [39] suggests that interaction is important in the development of shared norms and commitment to a political cause. By using a longitudinal design to examine the interaction between activists and non-activists, future research can illuminate the mechanisms that may reduce the psychological distance between the two groups and foster the creation of a strong and unified movement.

Furthermore, our studies did not examine whether communicating affective motivation (i.e., anger at the unjust situation or lack thereof) influences non-activists' and activists' mutual perceptions. Emotions like anger and moral outrage were found to be the key drivers of participation in collective action [10, 11]. Nevertheless, communicating anger as a reason to participate in collective action is not likely to be perceived positively by non-activists. Anger may be interpreted as a signal of aggressive behavior, and such negative stereotypes lead to less support for activists' cause among the general public [24]. On the other hand, a non-activist communicating lack of anger as a reason for inaction may be disliked by activists, because activists may interpret lack of anger as siding with the perpetrators. Future research should examine the impact of affective language on the relations between activists and their audiences.

## Conclusion

The present research fills an important gap in the literature on collective action by looking at the relations between those who engage in active struggle against injustice and those who do not. We showed that the motives put forward as explanations for (in)action play an important role in bringing or further separating the two groups. Our findings have important theoretical and practical implications, as they pointed out that activists may potentially mobilize more support for their cause if they reduce the distance they feel towards those who do not take part in collective action.

## Supporting information

**S1 Appendix. Activists' and non-activists' perceptions of the issues in Study1 and Study 2.**
(DOCX)

**S2 Appendix. Activists' perceptions of the COP21 protest in Study 3.**
(DOCX)

**S3 Appendix. Activists' and non-activists' perceptions of the anti-nuclear weapon protest in Study 4.**
(DOCX)

**S4 Appendix. The analyses with full sample in Study 4.**
(DOCX)

**S1 Table. Differences between activists and non-activists in Study 4.**
(DOCX)

**S2 Table. Study 4: Activists and general public views of both groups with full sample.**
(DOCX)

## Author Contributions

**Conceptualization:** Maja Kutlaca, Martijn van Zomeren, Kai Epstude.

**Formal analysis:** Maja Kutlaca.

**Methodology:** Maja Kutlaca.

**Supervision:** Martijn van Zomeren, Kai Epstude.

**Writing – original draft:** Maja Kutlaca.

**Writing – review & editing:** Maja Kutlaca, Martijn van Zomeren, Kai Epstude.

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
