## [Decision Letter · Decision Letter 0]

4 Nov 2019

PONE-D-19-25808

Friends or foes: How activists and non-activists perceive and evaluate each other

PLOS ONE

Dear Dr. Kutlaca,

Thank you for submitting your manuscript to PLOS ONE. After careful consideration, we feel that it has merit but does not fully meet PLOS ONE’s publication criteria as it currently stands. Therefore, we invite you to submit a revised version of the manuscript that addresses the points raised during the review process.

In addition to having independently read the manuscript, I have now received comments from two reviewers with expertise in the field of activism. As you can see below, both reviewers had positive things to say about your paper and differed primarily in the number of revisions needed for the paper to be publishable. In terms of my own reading of the paper, my opinion is a bit more in line with the comments from Reviewer 2. Specifically, I agreed that the theoretical basis for the hypotheses could be made clearer. I think clarifying the theoretical rationale would also make the hypotheses themselves easier to understand. Both reviewers also raised concerns about the samples of activists and non-activists that were collected. Reviewer 1 questions the extent to which we can be sure that the activist samples are in fact "activist" given that their inclusion is only based on one instance of activist activity. Reviewer 2 raised concerns about whether some of the comparisons between the activists and non-activists are valid (particularly in study 4). In terms of my own view, I shared these concerns about the samples used and also thought that some discussion is needed about the small size of some of the activist samples such as in study 4. To be clear, I don't think any of the issues raised either by myself or the reviewers are in any way insurmountable. 

We would appreciate receiving your revised manuscript by Dec 19 2019 11:59PM. To enhance the reproducibility of your results, we recommend that if applicable you deposit your laboratory protocols in protocols.io, where a protocol can be assigned its own identifier (DOI) such that it can be cited independently in the future. For instructions see: http://journals.plos.org/plosone/s/submission-guidelines#loc-laboratory-protocols

We look forward to receiving your revised manuscript.

Kind regards,

Daniel Wisneski

Academic Editor

PLOS ONE

Journal Requirements:

3. Thank you for including your ethics statement: University of Groningen

Study 1) ppo-014-054

Study 2)14100-N

Study 3) ppo-015-060

Study 4 ethics was obtained from the University of Osnabruck

Please amend your current ethics statement to confirm that your named institutional review board or ethics committee specifically approved this study.

4. Please ensure that you include a title page within your main document. You should list all authors and all affiliations as per our author instructions and clearly indicate the corresponding author.

Reviewers' comments:

Reviewer's Responses to Questions

**Comments to the Author**

1. Is the manuscript technically sound, and do the data support the conclusions?

Reviewer #1: Yes

Reviewer #2: Partly

2. Has the statistical analysis been performed appropriately and rigorously? 

Reviewer #1: Yes

Reviewer #2: I Don't Know

3. Have the authors made all data underlying the findings in their manuscript fully available?

Reviewer #1: Yes

Reviewer #2: Yes

4. Is the manuscript presented in an intelligible fashion and written in standard English?

Reviewer #1: Yes

Reviewer #2: Yes

5. Review Comments to the Author

Reviewer #1: This paper presents 4 studies examining how activists and non-activists view one another. The contribution is timely and novel. Very little work has examined these kinds of intergroup judgments, which are interesting both theoretically and practically. On the whole, the researchers seem to find that activists distance from non-activists more than non-activists distance from activists. This is particularly the case when non-activists deny moral responsibility for taking action. This asymmetry does have implications for how activists might engage (or not engage) non-activists in supporting the cause, potentially limiting the growth of social movements. It’s a nice set of studies, nice samples (of people engaging in collective action), and it fills a gap. I recommend the paper be published, after addressing the following concerns.

The patterns observed here are almost inevitably contingent on a whole range of factors—what is the ideology of the activists? Who is the target of activism? What is it’s goal? How do non-activists stand to benefit (or suffer) if the goal is achieved? How large is the movement? Is it radical or supported by large majorities of non-activists? For that reason, I am a concerned that about the way the paper presents activists and non-activists as “one-size-fits-all” labels, and suggests that the patterns observed here are generalizeable across activist contexts. To me, it seems more appropriate to consider how these evaluations emerge out of particular kinds of activist contexts. I think this general issue need to be raised and addressed in the intro, and in the general discussion, with particular reference to the claims about how activists perceive non-activists.

I think the authors also need to be a bit more careful with claims about how activists’ perceptions of non-activist might undermine the growth of the movement. That might happen, but we can’t actually be sure that the perceptions observed here actually DO result in behaviours by activists, or responses from non-activists that actually deter recruitment or more passive support. It might be worth qualifying those claims a bit more.

There is the implication that activists might have biased perceptions of non-activists, at list compared to the reverse. Of course, it might be that activists have perceptions that reflect a meaningful social reality, while non-activists bias there perceptions in a way that protects a superordinate or personal identity. Another way to approach the practical implications of the data is to suggest that activists might not fully tap into the support they might get from non-activists, given the rather positive impressions non-activists (sometimes) have of activists.

The activists samples are all based on people who participated in an action. Is it fair to characterize that sample as an activist sample? To what degree do we know that participants identified themselves as activists? If this is the only action they ever take, are they activists? I think the authors need to address the limitations of their samples in this regard, and the degree to which the activist label fits, and how this might limit generalizeability to other operationalizations of “activist”.

In the concluding paragraph, the authors suggest that activists might distance from non-activists to such a degree as to “redefine group boundaries”. What group are we talking about here, and where is the evidence for that?

Reviewer #2: How activists and non-activists perceive each other is a potentially interesting line of enquiry. However, the design and theorising around these studies needs to be clearer to shed any light. Following are areas that need most focus:

1. The hypotheses are not always clear (eg., H2b) and appear to be based more on assumptions than engagement with theory and evidence. For instance in relation to H1b it could equally be argued that in some interactional contexts activists might place a premium on individual responsibility to act (if you don't act who will). It would be good to see more consideration of counter-arguments and evidence as well as engagement with literature that does pertain to the relationship between activists and non-activists (e.g., around politicisation and movement building). Activists and non-activists are treated here as static de-contextualised categories which limits theorisation of the intergroup relationship.

2. Studies 1 and 2 show few significant effects and certainly don't warrant the strong claims made in this paper. The authors might want to consider the similarity of the two samples (contrary to the authors' assertions, participants in Study 1 were not Left -- they were at the centre of the scale and would more accurately be described as moderate - the description given to participants in Study 2 who were also towards the centre of the scale). Moreover, rather than very different populations, the activists were predominantly novices. What was most different was the context -- Study 1 was conducted at a protest where a shared identity would be salient.

3. The comparison of samples in Study 4 was equally problematic but for different reasons: 22 WILF activists compared with MTurk participants who were arguably unlikely to be part of WILF's movement potential (mostly men and no indication of their views on the activist issue) and who did not belong to a group that could provide the basis for self-categorisation.

4. Related to the above two points, the asymmetrical effects that have been found in the research could be explained in terms of group processes of differentiation on salient dimensions arising from self-categorisation for those at a protest (Study 1) and those belonging to a well defined activist group (Study 4).

5. The suggestion that this research shows that social change may depend on the relations between activists and non-activists is not warranted as it was not tested (top p.29); the suggestion that inclusion in the activist category requires that one participate in collective action (be an activist) seems tautological (p.29); and the claim that activists differentiated more clearly between non-activists in Study 3 compared to Study 1 is problematic given that they were only given the opportunity to evaluate different types of non-activist in Study 3.

6. Given that activist groups spend so much time and effort trying to understand non-activists for the purposes of recruitment and mobilization (certainly more than the latter spends trying to understand them) it seems naïve to conclude that what is needed is for activists to simply see non-activists as more like them. Surely the fact of recruitment and mobilization shows that matters might be more dynamic and complex. Whilst this research was not designed to consider real world complexity it would be good to see some reflection on this and the limits to validity of the findings.

6. PLOS authors have the option to publish the peer review history of their article (what does this mean?). If published, this will include your full peer review and any attached files.

Reviewer #1: No

Reviewer #2: No

---

## [Author Response · Author response to Decision Letter 0]

3 Feb 2020

Dear Dr. Wisneski, 

Thank you so much for giving us the opportunity to revise the manuscript and my sincere apologies for the delay with revising the manuscript. We agreed with the very helpful feedback and thus followed your and the reviewers’ suggestions. 

This means that, in the revised version of the manuscript, we included a more thorough discussion of the theoretical basis for our hypotheses in the General Introduction. Furthermore, we completely agree with your and the reviewers’ concerns about the sample limitations in our studies and we address this issue by a) providing a clearer definition of who activists and non-activists are in the General Introduction and more consistent operationalization of the two groups b) reflecting on how different contexts/samples of activists may lead to opposing results in the introduction to Study 3; and c) highlighting the limitations of our studies in the General Discussion. 

Below, we address each of the concerns raised by the reviewers in more detail. 

Yours Sincerely,

Maja Kutlaca, Martijn van Zomeren & Kai Epstude 

Reviewer #1: 

1) The patterns observed here are almost inevitably contingent on a whole range of factors—what is the ideology of the activists? Who is the target of activism? What is it’s goal? How do non-activists stand to benefit (or suffer) if the goal is achieved? How large is the movement? Is it radical or supported by large majorities of non-activists? For that reason, I am a concerned that about the way the paper presents activists and non-activists as “one-size-fits-all” labels, and suggests that the patterns observed here are generalizeable across activist contexts. To me, it seems more appropriate to consider how these evaluations emerge out of particular kinds of activist contexts. I think this general issue need to be raised and addressed in the intro, and in the general discussion, with particular reference to the claims about how activists perceive non-activists.

Response: We agree with the reviewer’s concerns and we reflect on this issue more specifically in the introduction of Study 3 (page 19-20). Prior research differentiates between specific contexts of activism such as structural vs. incidental disadvantages vs. issue-based activism (e.g., Curtin& McGarty, 2016), which are characterized by more or less established group identities (and associated group norms). The presence vs. absence of these identities may lead to more or less negative perceptions of non-activists. However, our findings are in line with prior research on relations between activists and the broader ingroup that may lack commitment to the cause (see Becker et al.,2011; Zaal et al., 2017): in our view, the negative evaluations of non-activists which we observe can be seen as complimentary strategy to disidentification from the broader ingroup found by prior research. 

2) I think the authors also need to be a bit more careful with claims about how activists’ perceptions of non-activist might undermine the growth of the movement. That might happen, but we can’t actually be sure that the perceptions observed here actually DO result in behaviours by activists, or responses from non-activists that actually deter recruitment or more passive support. It might be worth qualifying those claims a bit more.

3) There is the implication that activists might have biased perceptions of non-activists, at list compared to the reverse. Of course, it might be that activists have perceptions that reflect a meaningful social reality, while non-activists bias there perceptions in a way that protects a superordinate or personal identity. Another way to approach the practical implications of the data is to suggest that activists might not fully tap into the support they might get from non-activists, given the rather positive impressions non-activists (sometimes) have of activists.

Response points 2&3 : We agree with the reviewer and we used the suggestion provided to now discuss in the General Discussion (pages 33-34) how the observed patterns may prevent activists from seeking the support from non-activists. We note that this conclusion also fits nicely with findings by Becker and colleagues (2011), as well as Zaal and colleagues (2017). 

4) The activists samples are all based on people who participated in an action. Is it fair to characterize that sample as an activist sample? To what degree do we know that participants identified themselves as activists? If this is the only action they ever take, are they activists? I think the authors need to address the limitations of their samples in this regard, and the degree to which the activist label fits, and how this might limit generalizeability to other operationalizations of “activist”.

Response: We agree with the reviewer that indeed one instance of participation in collective action cannot be fully equated with activism. We now include a better definition of activism in the General Introduction (see pages 4-5). More specifically we use participation in collective action and membership in a political organization as criteria to differentiate between activists and non-activists (this is also in line with the existing literature, see Curtin & McGarty, 2016). Thus we included both novices and experienced activists, and we avoided the issues with the activist label (Drury et al., 2003), and the issues with defining activists according to their motivations, because they can also be context dependent. 

Importantly, the limitation pointed out by the Reviewer only applies to the sample of Study 1, whereas the samples in Study 3 and Study 4 consisted of predominantly experienced activists who identified as long-term members of various social movements (for more details please see method section in Study 3 and Study 4 – we added this information in Study 4, which was missing). 

5) In the concluding paragraph, the authors suggest that activists might distance from non-activists to such a degree as to “redefine group boundaries”. What group are we talking about here, and where is the evidence for that?

Response: We removed these too strong claims and now we discuss our findings in light of the support the activists may fail to garner to the fullest as suggested by the reviewer. 

Reviewer #2: 

1. The hypotheses are not always clear (eg., H2b) and appear to be based more on assumptions than engagement with theory and evidence. For instance in relation to H1b it could equally be argued that in some interactional contexts activists might place a premium on individual responsibility to act (if you don't act who will). It would be good to see more consideration of counter-arguments and evidence as well as engagement with literature that does pertain to the relationship between activists and non-activists (e.g., around politicisation and movement building). Activists and non-activists are treated here as static de-contextualised categories which limits theorisation of the intergroup relationship.

Response: We completely agree with the reviewer and we now include a thorough discussion of our hypotheses (see pages 6-7 in the General Introduction). We also acknowledge opposing predictions as suggested by the reviewer: personal responsibility is particularly important in the context of environmental activism, which is the case in Study 3. Lastly, we include the limitation of our cross-sectional designs which artificially reduces the definition of activists and non-activists to one instance of participation in a collective action (see Limitations on page 34). 

2. Studies 1 and 2 show few significant effects and certainly don't warrant the strong claims made in this paper. The authors might want to consider the similarity of the two samples (contrary to the authors' assertions, participants in Study 1 were not Left -- they were at the centre of the scale and would more accurately be described as moderate - the description given to participants in Study 2 who were also towards the centre of the scale). Moreover, rather than very different populations, the activists were predominantly novices. What was most different was the context -- Study 1 was conducted at a protest where a shared identity would be salient.

Response: We agree that it is wise to tone down our conclusions more. We now refer to the suggestion made by Reviewer 1 about activists’ failing to fully tap into the support they might get from non-activists (see pages 33-34 in the General Discussion). We corrected the political orientation in Study 1. We note that although the sample in Study 1 consisted of novices and inexperienced activists, this was not the case with samples in Study 3 (pages 20-21) and Study 4 (page 28 – we added the information that was missing originally). Moreover, the findings of Study 1 were replicated in Study 3, which was conducted after the protest has already taken place (rather than on-site). This suggests that the context of protest may not be problematic for the interpretation of our findings. 

3. The comparison of samples in Study 4 was equally problematic but for different reasons: 22 WILF activists compared with MTurk participants who were arguably unlikely to be part of WILF's movement potential (mostly men and no indication of their views on the activist issue) and who did not belong to a group that could provide the basis for self-categorisation.

Response: We agree with Reviewer’s concern and we mention the limitation of Study 4 more explicitly (see page 32). We included the questions about the extent to which both activist and non-activists in all studies endorsed the issue and identified with the group (we included this information in the Supplementary Materials). In Study 4 (but also in Study 2), the non-activist sample overall agreed with and supported the activists’ cause: the means were all larger than the medium scale point (please see Table S1 in the Supplementary Materials) therefore, they could have been part of the movement potential. To be consistent with our definitions of activists and non-activists, we excluded those M-turk participants who indicated that they were already politically active. Still, the sample now used in the analyses was relatively positive about the activists’ cause. 

Moreover, in the experimental studies (Study 1 & 3) the manipulations explicitly said that a non-activist agreed with the activists’ cause to ensure that the non-activist could be considered as part of the movement’s mobilization potential. Therefore, the reason for negative perceptions could not be due to disagreement in opinions between activists and non-activists, but their reason not to act. In Study 4, we only mentioned those who were not there, we did not specify that they would disagree with the activists’ cause. 

4. Related to the above two points, the asymmetrical effects that have been found in the research could be explained in terms of group processes of differentiation on salient dimensions arising from self-categorisation for those at a protest (Study 1) and those belonging to a well defined activist group (Study 4).

Response: We agree with the reviewer that this may be the case and also mentioned in the General Discussion that one explanation for activists’ responses to non-activists may be derived from their need for positive distinctiveness (please see pages 33-34). 

5. The suggestion that this research shows that social change may depend on the relations between activists and non-activists is not warranted as it was not tested (top p.29); the suggestion that inclusion in the activist category requires that one participate in collective action (be an activist) seems tautological (p.29); and the claim that activists differentiated more clearly between non-activists in Study 3 compared to Study 1 is problematic given that they were only given the opportunity to evaluate different types of non-activist in Study 3.

Response: We deleted these conclusions and we only discuss how our findings indicate that activists may not fully tap into the support they might get. Regarding the comparison between Study 1 and Study 3 – in both studies the activists were asked to evaluate different types of non-activists. The design of Study 3 included an additional condition, i.e., engagement in individual action as the reason not to take part in the collective action. Moreover, we cite prior research that found that more seasoned activists care more about the movement building than the novices (Blackwood & Louis, 2012), which indeed might make them less judgmental towards non-activists. Thus, instead of discussing how experienced activists may be better at differentiating between different types of non-activists, we propose that they may indeed have more positive views and feel less personally distant. However, our findings do not necessarily support this conclusion: although the activists generally evaluated the non-activists who engaged in individual action more positively, similarly to Study 1 they did not feel personally close to non-activists irrespective of the reason they communication for inaction. 

6. Given that activist groups spend so much time and effort trying to understand non-activists for the purposes of recruitment and mobilization (certainly more than the latter spends trying to understand them) it seems naïve to conclude that what is needed is for activists to simply see non-activists as more like them. Surely the fact of recruitment and mobilization shows that matters might be more dynamic and complex. Whilst this research was not designed to consider real world complexity it would be good to see some reflection on this and the limits to validity of the findings.

Response: We agree with the reviewer and we included this limitation in the General Discussion (see page 34).

---

## [Editor Report · Decision Letter 1]

18 Feb 2020

PONE-D-19-25808R1

Friends or foes? How activists and non-activists perceive and evaluate each other

PLOS ONE

Dear Dr. Kutlaca,

Thank you for submitting your manuscript to PLOS ONE. After careful consideration, we feel that it has merit but does not fully meet PLOS ONE’s publication criteria as it currently stands. Therefore, we invite you to submit a revised version of the manuscript that addresses the points raised during the review process.

I have now carefully read both your responses to the reviewers' comments and your revised manuscript. First, I would like to thank you for the care you clearly took in addressing the changes both I and the reviewers thought were warranted. I think the current manuscript is stronger and clearly exceeds the bar for publication in PLOS ONE. That said, I do think that the paper would benefit from one more minor change. Specifically, I still think that the description of the hypotheses in the introduction could be clarified. As it is currently written, the the distinction between the moral vs. instrumental justifications is clear, but I think readers will still have trouble following the  distinction between the individual vs. collective reasons for action/inaction. The opening sentence for the second paragraph on page 6, for example, does little to set up how hypotheses 1a and 1b differ. Given the importance of this section of the introduction for the rest of the paper, I think a little more could be done to make it clearer exactly how the different parts of the hypotheses (i.e., 1a vs 1b and 2a vs 2b) differ. Again, I know this is a very minor revision, but I think it would be beneficial. 

We would appreciate receiving your revised manuscript by Apr 03 2020 11:59PM. To enhance the reproducibility of your results, we recommend that if applicable you deposit your laboratory protocols in protocols.io, where a protocol can be assigned its own identifier (DOI) such that it can be cited independently in the future. For instructions see: http://journals.plos.org/plosone/s/submission-guidelines#loc-laboratory-protocols

We look forward to receiving your revised manuscript.

Kind regards,

Daniel Wisneski

Academic Editor

PLOS ONE

---

## [Author Response · Author response to Decision Letter 1]

10 Mar 2020

Dear Dr. Wisneski, 

Thank you again for giving us the opportunity to revise the manuscript. We hope we addressed your concerns regarding the theoretical distinctions between the hypotheses by providing clearer theoretical arguments for each one of them (see pages 6-8). Moreover, we felt our labeling of hypotheses as H1a & H1b for instance added to the confusion, because they could be interpreted as competing predictions, which is not the case here (therefore we relabeled them as Hypothesis 1 to 5). Moreover, the limitation section (page 35) now addresses another potential motivation for (in)action, namely emotions, which was not included in our research. Thus, we hope our manuscript is now better grounded in the social-psychological literature on collective action and will be of interest to the readership. 

Thank you for your time and consideration. We look forward to hearing from you at the earliest time of your convenience.

Yours Sincerely,

Maja Kutlaca, Martijn van Zomeren & Kai Epstude

---

## [Editor Report · Decision Letter 2]

12 Mar 2020

Friends or foes? How activists and non-activists perceive and evaluate each other

PONE-D-19-25808R2

Dear Dr. Kutlaca,

We are pleased to inform you that your manuscript has been judged scientifically suitable for publication and will be formally accepted for publication once it complies with all outstanding technical requirements.

With kind regards,

Daniel Wisneski

Academic Editor

PLOS ONE
---

## [Editor Report · Acceptance letter]

25 Mar 2020

PONE-D-19-25808R2 

Friends or foes? How activists and non-activists perceive and evaluate each other 

Dear Dr. Kutlaca:

I am pleased to inform you that your manuscript has been deemed suitable for publication in PLOS ONE. Congratulations! Your manuscript is now with our production department. 

With kind regards,

on behalf of

Dr. Daniel Wisneski 

Academic Editor

PLOS ONE